# Uptake of COVID-19 Vaccination and Its Associated Factors among College Students in China: A Nationwide Cross-Sectional Study

**DOI:** 10.3390/ijerph20042951

**Published:** 2023-02-08

**Authors:** Xuelian Xu, Junye Bian, Zhihui Guo, Xinyi Li, Weijie Zhang, Bingyi Wang, Yinghui Sun, Xiaojun Meng, Huachun Zou

**Affiliations:** 1School of Literature and Education, Bengbu University, Bengbu 233030, China; 2School of Public Health (Shenzhen), Sun Yat-sen University, Shenzhen 518107, China; 3Wuxi Municipal Center for Disease Control and Prevention, Wuxi 214023, China

**Keywords:** COVID-19, vaccination, college students, education, China

## Abstract

Our study aims to assess the uptake of COVID-19 vaccination and its associated factors among Chinese college students. A web-based cross-sectional study was conducted from 18 May to 17 June 2022. A total of 3916 participants were included. The coverage of the first dose, complete vaccination and booster vaccination among college students was 99.49%, 81.96% and 79.25%, respectively. College students with an older age (AOR: 0.72, 95% CI: 0.57–0.90), non-medical major (0.47, 0.37–0.61) and studying in north-east China (0.35, 0.22–0.58) were less likely to complete vaccination. Individuals who were female (1.62, 1.35–1.94) and received a recombinant subunit vaccine (8.05, 5.21–12.45) were more likely to complete vaccination. Non-medical students (0.56, 0.43–0.73) and students studying in north-east China (0.28, 0.16–0.49) were less likely to receive a booster dose, while female students (1.51, 1.23–1.85) had a higher likelihood. The main reason for being unvaccinated was “contraindication” (75.00%), and the main reason for not receiving a booster dose was “being too busy to attend to it” (61.37%). This study demonstrated a high adherence to the COVID-19 vaccination policy among Chinese college students. Targeted strategies should be applied to remove barriers to COVID-19 vaccination among college students.

## 1. Introduction

The coronavirus disease 2019 (COVID-19), a respiratory syndrome caused by SARS-CoV-2, has brought about a worldwide pandemic since its breakout in 2019. As of 9 November 2022, there were over 600 million confirmed cases of COVID-19, including nearly 7 million deaths globally. In China, these figures were over 9 million and nearly 29 thousand [1], respectively.

COVID-19 cases in young adults, including college students, are more likely to be asymptomatic or paucisymptomatic carriers of SARS-CoV-2 compared to their older counterparts [2]. As the new variant spreads faster, if a case is not identified in time, the inevitable daily close contact may result in clustered epidemics and large-scale transmission [3]. In addition to the risk of infection, overwhelming evidence has shown that continued widespread disruption to the education system due to COVID-19 seriously threatened both the physical and mental health of college students [4,5,6]. As a result, both the WHO and UNICEF underscored the importance of decisive action to reduce the in-school transmission of COVID-19 [7]. In order to better protect this population, the Chinese Ministry of Education and the National Health Commission jointly issued the Technical Plan for the Prevention and Control of COVID-19 in the Spring semester for colleges (the Third Edition) in order to ensure COVID-19 vaccination among college students [8].

Three types of COVID-19 vaccines are available in China, including the single-dose adenovirus vector vaccine, the 2-dose inactivated vaccine and the 3-dose recombinant subunit vaccine. However, with the emergence of Omicron variant (B.1.1.529), full vaccination may not be enough to prevent infection. Receiving a booster dose is of great importance in the context of the Omicron wave [9,10]. To address this issue, China called on key populations who completed the primary vaccination series of the adenovirus vector vaccine and inactivated vaccine to receive a booster dose [11].

Previous studies found that the willingness to get COVID-19 vaccination was high among college students, but vaccine hesitancy was also common [12,13,14]. Nearly one-third of college students were hesitant or resistant to get vaccinated, significantly more than their older counterparts [15]. Previous studies found that vaccine hesitancy among college students was associated with socio-demographics such as gender, age [15], college location [13] and major [12]. Concerns about side effects and effectiveness also contribute to hesitancy [15]. A few studies reported college students’ actual vaccination status with the influence of vaccine hesitancy. Xiong et al. found that nearly half of college students with vaccine hesitancy would still get vaccinated [12], which may partially account for the high coverage (over 95%) of first-dose vaccination among them [16]. Cao [17] and Li [18] found that the uptake of COVID-19 vaccination among college students was also associated with gender, age and major, which was similar to vaccine hesitancy. However, with the promotion of the COVID-19 booster vaccination, vaccine hesitancy may increase significantly [19]. Existing studies in China only reported data from before the promotion of the booster vaccination and had smaller samples that were recruited from a single site. The latest uptake of COVID-19 vaccination among college students is unclear. Therefore, we aimed to conduct a nationwide survey to assess the uptake of COVID-19 vaccination and its associated factors among Chinese college students after the promotion of the booster dose.

## 2. Materials and Methods

### 2.1. Study Design

From 18 May to 17 June 2022, we conducted an online survey on college students in China. A questionnaire was set up via “Wenjuanxing”, a Chinese online platform providing survey functions. A poster of our survey was distributed via WeChat and QQ (two of the most popular instant messaging platforms in China). By scanning the QR code on the posters, college students who were interested in our research could access the online questionnaire.

### 2.2. Sample Size

Given an alpha value of 0.05 and the permissible error δ of 0.02, and assuming the proportion of college students already having received a COVID-19 booster dose was 60%, a minimum of 2305 subjects were required to provide sufficient statistical power [20].

### 2.3. Participants

Individuals who met the following criteria were eligible in this study: (1) college students (junior college students, undergraduate students and graduate students; and (2) individuals aged 16 years and older but younger than 35 years old. By checking the IP address, questionnaires from countries and regions other than the Chinese mainland were excluded. Questionnaires with an illogical answer to the question “Why haven’t you completed the vaccination?” were regarded as being invalid.

### 2.4. Measures

The following information was self-reported by participants: (1) socio-demographic information, including gender (male/female), age (in years), ethnicity (Han/others), college location (all cities in China were displayed as options), education (junior college student or undergraduate student/graduate student), major (medical/non-medical) and family monthly income (<10,000/10,000–20,000/>20,000); (2) COVID-19 vaccination status (unvaccinated/vaccinated); (3) reasons for not receiving a first-dose and booster vaccination (participants could choose from at least one of the following options: contraindications/concerns about effectiveness/fear of adverse reactions/being too busy to attend to it/I think it is unnecessary/inconvenience; a blank was left to fill in for any other reason not mentioned above) were also collected.

Those who received at least one dose of a COVID-19 vaccine were required to offer detailed information about the received vaccine, including its production company and the number of received doses. According to this, the COVID-19 vaccination status among vaccinated individuals was further classified. A full COVID-19 vaccination referred to receiving one dose for the adenovirus vector vaccine, two doses for the inactivated vaccine and three doses for the recombinant subunit vaccine. Since the booster dose was only required for people who received the adenovirus vector vaccine and inactivated vaccine in China, a complete COVID-19 vaccination referred to getting a full COVID-19 vaccination with the booster dose for the adenovirus vector vaccine or the inactivated vaccine, or getting a full COVID-19 vaccination for the recombinant subunit vaccine.

### 2.5. Outcomes

The primary outcome was the completion of COVID-19 vaccination, which reflected college students’ adherence to vaccination guidelines. All vaccinated participants were included and classified according to the COVID-19 vaccination status.

The secondary outcome was the uptake of COVID-19 booster dose. The analysis was based on participants who got a full vaccination with the adenovirus vector vaccine and inactivated vaccine. They were classified as fully vaccinated with a booster dose and fully vaccinated without a booster dose according to the uptake of the booster dose. Exploratory outcomes included reasons for not getting vaccinated or not receiving the booster dose.

The coverage was reported from three dimensions, including first-dose vaccination, full vaccination and booster vaccination.

### 2.6. Statistical Analysis

In the descriptive analyses, characteristics of the study participants were expressed using frequencies and percentages. A chi-square test was applied for bivariate analyses. Logistic regression was used to further examine factors associated with complete COVID-19 vaccination and the uptake of the booster dose. Variables with a *p*-value less than 0.10 in the bivariate analyses were introduced in multivariable logistic regression models. All effects were estimated with 95% CI and *p*-values. Statistical significance was taken as two-sided *p*-values < 0.05. All analyses were performed using SPSS 20.0 (IBM Corporation, New York, NY, USA) and Excel 2010 (Microsoft, Redmond, WA, USA).

### 2.7. Ethics Approval

This research was approved by the Ethical Review Committee of School of Public Health (Shenzhen), Sun Yat-sen University (SYSU-PHS-2022025). Electronic informed consent was obtained from all participants before commencing the survey. Participants who completed the questionnaire entered a lottery to receive up to 50 CNY (approximately 7 USD) of reimbursement.

## 3. Results

### 3.1. Characteristics of the Participants

In general, 4613 questionnaires were collected over the study period. After excluding invalid questionnaires, a total of 3916 participants were included in the study (Figure 1). Table 1 summarizes the characteristics of the sample. More than half of the respondents were female (64.56%), 83.09% were 23 years and younger, only 9.91% were graduate students, and 22.06% majored in medicine. The coverage of first-dose vaccination was 99.49%. A total of 75.00% of the 20 unvaccinated students did not receive at least one dose of vaccine because of contraindication, which was the most common reason (Figure 2).

### 3.2. Factors Associated with Complete COVID-19 Vaccination

Table 2 shows the completion of COVID-19 vaccination among the respondents who got vaccinated. The coverage of complete vaccination was 81.96%. Gender, age, college location, major and the type of received vaccine were associated with a complete COVID-19 vaccination. Lower odds of complete vaccination were found in students aged 24 and older (OR: 0.72, 95% CI: 0.57–0.90), students studying in north-east China (OR: 0.35, 95% CI: 0.22–0.58) and non-medical students (OR: 0.47, 95% CI: 0.37–0.61). Being female (OR: 1.62, 95% CI: 1.35–1.94) and receiving a recombinant subunit vaccine (OR: 8.05, 95% CI: 5.21–12.45) were associated with higher odds of complete vaccination.

### 3.3. Factors Associated with Uptake of COVID-19 Booster

Among the respondents who got vaccinated, 2607 students got a full vaccination with the adenovirus vector vaccine or inactivated vaccine, and the coverage of the booster vaccination was 79.25%. As shown in Table 3, bivariate analysis indicated the association of gender, major, and college location with the uptake of the COVID-19 booster dose. Students who received different types of vaccine did not show significant differences in the uptake of the COVID-19 booster dose. Non-medical students (OR: 0.56, 95% CI: 0.43–0.73) and students studying in north-east China (OR: 0.28, 95% CI: 0.16–0.49) had lower odds of receiving a COVID-19 booster dose, while female students (OR: 1.51, 95% CI: 1.23–1.85) had higher odds. For those who got a full course of vaccination but did not receive a booster dose, “being too busy to attend to it” was the most common reason (Figure 3).

## 4. Discussion

To the best of our knowledge, this is the first nationwide study on the COVID-19 vaccination status and associated factors among Chinese college students that included booster vaccination. Our study found that the coverage of first-dose vaccination, complete vaccination and booster vaccination among the respondents was 99.49%, 81.96% and 79.25%, respectively, as of June 2022. A complete COVID-19 vaccination and uptake of the booster dose was associated with gender, major and college location.

We found that the coverage of first-dose vaccination among the respondents was 99.49%, which is consistent with the data released by the Chinese government (over 95% among students aged 18 years and older) [16]. With the promotion of COVID-19 vaccination, the most common reason for being unvaccinated changed into “contraindications”, instead of “concerns about vaccines (such as the side effect and effectiveness of vaccine)” in a previous study [15]. As for the booster dose, compared with the coverage of 69.07% among all populations as of 12 October 2022, the coverage of the booster dose among college students in this study was relatively high [21]. This may be attributed to their good health literacy and high educational level [22]. The frequent use of social media and various channels for acquiring knowledge have enabled them to have better knowledge and attitudes towards COVID-19 vaccines and to put this into practice [23]. The successful experience of COVID-19 vaccination among college students can be extended to vaccination among other populations or against other pathogens.

Previous studies have shown that female students and medical students had better knowledge and acceptability towards COVID-19 vaccination [12,15,24,25,26]. We found that such gender and major differences also existed in the practice of COVID-19 vaccination and in the adherence to its guideline. Thus, all these studies demonstrated that precision health education should be implemented among college students, and attention must be focused on male students, particularly non-medical students. It was also reported that recommendations from surrounding medical professionals or people who have been vaccinated are important factors in promoting vaccination [27,28]. Therefore, colleges may invite experts or doctors to disseminate knowledge about COVID-19 vaccination. Simultaneously, we should also consider positive peer effects. Herd mentality could drive students with COVID-19 vaccine hesitancy into taking it [12]. Students who have completed the vaccination have the potential to motivate others to better comply with vaccination policies.

Age is another factor associated with college students’ COVID-19 vaccination. Younger students were more likely to complete the COVID-19 vaccination. Previous studies showed that vaccine hesitancy was more prevalent among senior students [29,30]. This may be because seniors face intense pressure to graduate, move on to higher education and find jobs. They are often too busy to have time to comply with multiple doses. This is also the most common reason for not receiving a booster dose in our study. Thus, colleges should set up on-campus vaccination sites and implement targeted vaccination strategies, especially with regard to students’ curriculum. However, such an age difference was not obvious in the uptake of the booster dose. It is necessary to strengthen the promotion of booster doses among college students of all age groups.

Moreover, similar to previous studies [12,15], the family-income level was not associated with vaccination behavior in our logistic regression model. Since COVID-19 vaccination is free of charge in China, concerns about vaccine prices would not be an objective barrier to vaccination for Chinese students. However, other objective impediments should also be taken into account when formulating policy. It is interesting to find that students in colleges located in north-east China had a significantly lower coverage of complete vaccination and booster vaccination. Such results may be related to the local epidemic situation. When booster vaccination was widely carried out, north-east China was one of the areas that suffered from the worst COVID-19 outbreak. For example, Changchun, one of the largest cities in north-east China, has been under strict management for more than a month [31]. Consequently, local colleges had no alternative but to lock down in order to prevent clustered epidemics on campus. The result of our study does remind us that school lockdowns during the epidemic may have a great impact on staged medication such as vaccination. Separate, well-supplied and professional health stations and vaccination sites should be established in schools for emergencies.

Another important result in our study is that complete vaccination was associated with the type of received vaccine. The coverage of complete vaccination for respondents who did not require a booster dose (individuals who received the recombinant subunit vaccine) was higher than that for those who required one (individuals who received the adenovirus vector vaccine or the inactivated vaccine). We also found that the coverage of booster vaccination was commonly suboptimal, regardless of the type of received vaccine. This is consistent with a previous study, according to which vaccine hesitancy increased significantly with the promotion of booster vaccination [19]. Lai et al. found that such a phenomenon could be explained by the low perception of a booster vaccination’s benefits [32]. Thus, these results indicated that comprehensive health education about a COVID-19 booster dose is of great importance.

This study has some limitations. First, this study was limited by its cross-sectional nature and could not establish causal relationships. Second, this study was a web-based survey based on convenience sampling, and selection bias was inevitable. Respondents in our study may be more concerned about information about COVID-19 vaccination and may therefore be more likely to complete vaccination. Third, we only focused on the differences in terms of a complete vaccination and an uptake of the booster dose between students with different socio-demographic characteristics, but the questionnaire may not be comprehensive enough to explain the underlying reasons. Finally, the vaccination status was self-reported by respondents, which may not be as accurate as clinical vaccination records. However, since personal vaccination records can be easily found via smartphone in China and the vaccination information was required to be reported in detail, information bias in this study is unlikely to be significant.

## 5. Conclusions

The current study demonstrated a high adherence to the COVID-19 vaccination policy among college students in China, but the coverage of the booster vaccination was suboptimal. Male students, non-medical students and students studying in north-east China were associated with a lower coverage of complete vaccination and booster vaccination. It is necessary to foster further systematic and continuous health education about COVID-19 vaccination on campus. Colleges should set up separate, well-supplied and professional health stations and vaccination sites for emergencies. Meanwhile, targeted vaccination strategies among college students should be applied in order to address barriers to COVID-19 vaccination, especially students’ curriculum schedules.

## Figures and Tables

**Figure 1 ijerph-20-02951-f001:**
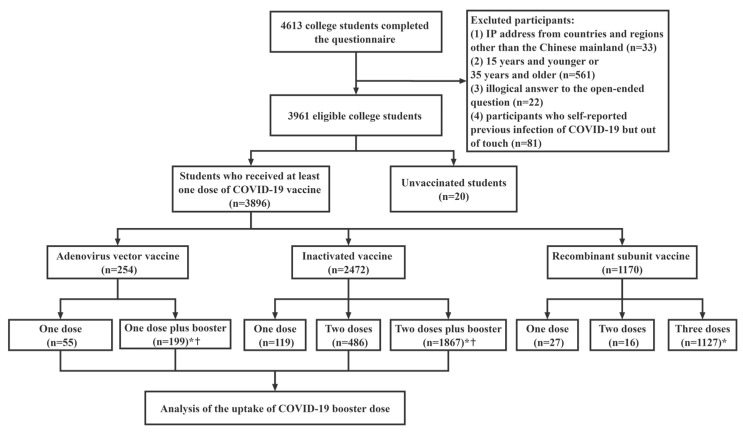
Flow chart of selection and analyses of the study respondents. * refers to the respondents who have completed vaccination. † refers to the respondents who have received a booster dose.

**Figure 2 ijerph-20-02951-f002:**
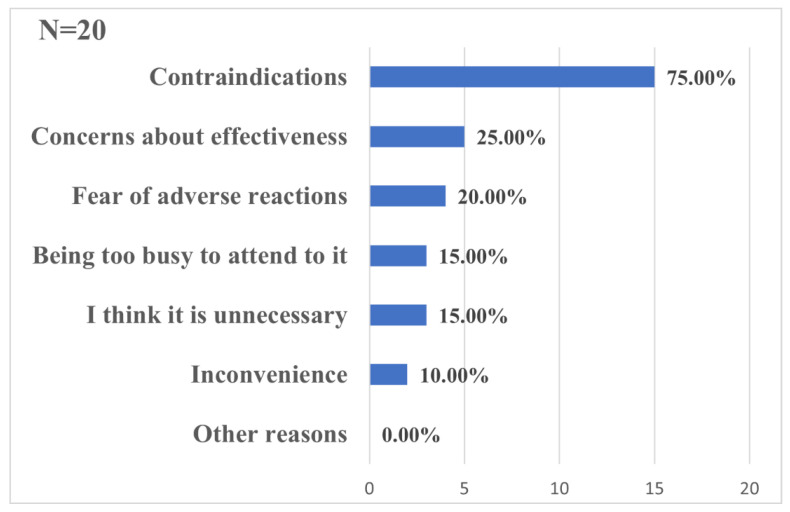
Reasons for not receiving the COVID-19 vaccine (%) among college students in China (N = 20).

**Figure 3 ijerph-20-02951-f003:**
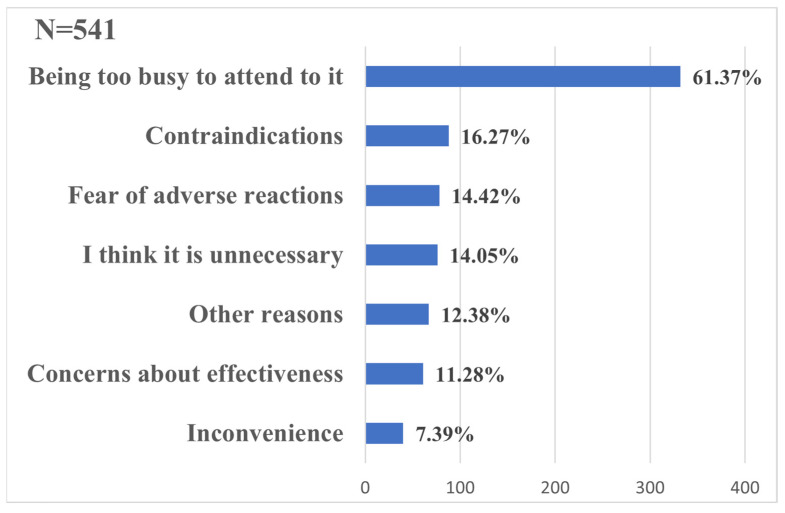
Reasons for not receiving a COVID-19 booster dose (%) among college students in China (N = 541).

**Table 1 ijerph-20-02951-t001:** Characteristics of respondents (N = 3916).

Characteristics	n	%
Sex		
Male	1388	35.44
Female	2528	64.56
Age (year)		
16–23	3254	83.09
≥24	662	16.91
Ethnicity		
Han	3618	92.39
Others	298	7.61
College location ^a^		
East	1269	32.41
Central	2137	54.57
West	371	9.47
North-east	139	3.55
Education		
Junior college student/ Undergraduate student	3528	90.09
Graduate student	388	9.91
Major		
Medical	864	22.06
Non-medical	3052	77.94
Family monthly income (CNY ^b^)		
<10,000	3140	80.18
10,000–20,000	639	16.32
>20,000	137	3.50
Vaccination		
Unvaccinated	20	0.51
Vaccinated	3896	99.49

Notes: ^a^ College location is divided according to The Division Method of East, West, Central and Northeast Regions issued by the National Bureau of Statistics of China in 2011. ^b^ 1 CNY ≈ 0.15 USD.

**Table 2 ijerph-20-02951-t002:** Completion of the COVID-19 vaccination among Chinese college students (N = 3896).

Characteristics	N	Bivariate Analysis			Multivariate Analysis		
		Incomplete Vaccination (%)	Complete Vaccination (%)	*p*	AOR	95% CI	*p*
Overall	3896	18.04	81.96				
Gender				<0.001 *			
Male	1377	23.17	76.83		Ref		
Female	2519	15.24	84.75		1.62	1.35–1.94	<0.001 *
Age (year)				<0.001 *			
16–23	3240	16.82	83.18		Ref		
≥24	656	24.09	75.91		0.72	0.57–0.90	0.004 *
Ethnicity				0.53			
Han	3602	17.93	82.07				
Others	294	19.39	80.61				
College location ^a^				<0.001 *			
West	369	13.01	86.99		Ref		
North-east	136	36.76	63.24		0.35	0.22–0.58	<0.001 *
East	1259	21.68	78.32		0.94	0.65–1.35	0.72
Central	2132	15.57	84.43		1.33	0.92–1.92	0.13
Education				0.14			
Junior college student/ Undergraduate student	3511	17.74	82.26				
Graduate student	385	20.78	79.22				
Major				<0.001 *			
Medical	862	13.46	86.54		Ref		
Non-medical	3034	19.35	80.65		0.47	0.37–0.61	<0.001 *
Family monthly income (CNY ^b^)				0.007 *			
<10,000	3126	17.08	82.92		Ref		
10,000–20,000	634	21.77	78.23		0.86	0.68–1.08	0.19
>20,000	136	22.79	77.21		0.82	0.53–1.28	0.39
Type of received vaccine				<0.001 *			
Adenovirus vector vaccine ^c^	254	21.65	78.35		Ref		
Inactivated vaccine ^d^	2472	24.47	75.53		0.84	0.61–1.16	0.29
Recombinant subunit vaccine ^e^	1170	3.68	96.32		8.05	5.21–12.45	<0.001 *

Notes: ^a^ College location is divided according to The Division Method of East, West, Central and Northeast Regions issued by the National Bureau of Statistics of China in 2011. ^b^ 1 CNY ≈ 0.15 USD. ^c^ People who receive adenovirus vector vaccine only have to take up one dose for full vaccination but are recommended to receive a booster dose. ^d^ People who receive inactivated vaccine have to take up two doses for full vaccination but are recommended to receive a booster dose. ^e^ People who receive recombinant subunit vaccine have to take up three doses for full vaccination but have not been recommended to receive a booster dose as of June 2022. * refers to characteristics of statistically difference.

**Table 3 ijerph-20-02951-t003:** Uptake of COVID-19 booster dose among Chinese college students who got full vaccination with the adenovirus vector vaccine or the inactivated vaccine (N = 2607).

Characteristics	N	Bivariate Analysis			Multivariate Analysis		
		without Booster (%)	with Booster (%)	*p*	AOR	95% CI	*p*
Overall	2607	20.75	79.25				
Gender				<0.001 *			
Male	854	26.46	73.54		Ref		
Female	1753	17.97	82.03		1.51	1.23–1.85	<0.001 *
Age (year)				0.005 *			
16–23	2203	19.79	80.21		Ref		
≥24	404	26.99	74.01		0.77	0.59–1.00	0.051
Ethnicity				0.56			
Han	2452	20.64	79.36				
Others	155	22.58	77.42				
College location ^a^				<0.001 *			
West	190	17.89	82.11		Ref		
North-east	99	43.43	56.57		0.28	0.16–0.49	<0.001 *
East	813	22.14	77.86		0.99	0.65–1.52	0.98
Central	1505	18.87	81.13		1.10	0.72–1.68	0.66
Education				0.30			
Junior college student/ Undergraduate student	2349	20.48	79.52				
Graduate student	258	23.26	76.74				
Major				0.004 *			
Medical	604	16.56	83.44		Ref		
Non-medical	2003	22.02	78.98		0.56	0.43–0.73	<0.001 *
Family monthly income (CNY ^b^)				0.01 *			
<10,000	2100	19.62	80.38		Ref		
10,000–20,000	414	24.88	75.12		0.81	0.63–1.05	0.12
>20,000	93	27.96	72.04		0.65	0.40–1.06	0.08
Type of received vaccine				0.71			
Adenovirus vector vaccine ^c^	254	21.65	78.35				
Inactivated vaccine ^d^	2353	20.65	79.34				

Notes: ^a^ College location is divided according to The Division Method of East, West, Central and Northeast Regions issued by the National Bureau of Statistics of China in 2011. ^b^ 1 CNY ≈ 0.15 USD. ^c^ People who receive adenovirus vector vaccine only have to take up one dose for full vaccination but are recommended to receive a booster dose. ^d^ People who receive inactivated vaccine have to take up two doses for full vaccination but are recommended to receive a booster dose. * refers to characteristics of statistically difference.

## Data Availability

Not applicable.

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
