# Peer review of "Uptake of COVID-19 Vaccination and Its Associated Factors among College Students in China: A Nationwide Cross-Sectional Study"

_ijerph, 2023, doi:10.3390/ijerph20042951_

Round 1
Reviewer 1 Report
In general, it is a well-articulated paper. The study findings have added value and I enjoyed readying it.
The study accessed the uptake of COVID-19 vaccination among college students in the real word. They estimated the coverage of first dose, complete vaccination and booster vaccination. They found lower coverage of complete vaccination and booster dose vaccination was associated with male students, non-medical students and students studying in North-east China. Also, this study found the most common reason for being unvaccinated and not receiving booster dose.
The participants were recruited nationwide through an online questionnaire, the sample size was considerable and uptake of booster dose was included in the study. These were the strengths of the study. However, online survey also faces the influence of selection bias due to its convenience sampling process.
Major revision:
1. The outcome measurement was not clearly stated in the manuscript. This was related to the validity of the findings. I think the authors should state this issue clearly in the “Methods” section and fully discuss its impact on the results.
2. Were there any previous studies reported the status of COVID-19 vaccination in college students and its correlate? What are the main correlates of vaccination hesitancy in these previous studies? Please state in the introduction.
Minor revision:
1. In “Abstract” section, when the format of the content in the first bracket is determined, the subsequent “AOR” and “95%CI” can be omitted.
2. Please note the format in “Characteristics of the participants” section (Line 122, Page3).
3. Please maintain the consistency of reporting all decimal places across the manuscript. P values should be expressed to 2 digits to the right of the decimal point unless the first 2 digits are zeros, in which case 3 digits to the right of the decimal place should be provided. However, values close to .05 may be reported to 3 decimal places because the .05 is an arbitrary cut point for statistical significance.
4. There should be no lines below the total row in Table 2 and Table 3, and the lines between bivariate and multivariate analysis should break.
Author Response
Thank you very much for giving us the opportunity to revise our manuscript. We sincerely appreciated the valuable comments and suggestions. We have carefully revised our manuscript according to your suggestions in a point-to-point manner with track changes. We believe that the quality of our manuscript has been substantially improved after addressing these edits. The main corrections and responses are as follows:
Major revision:
1. The outcome measurement was not clearly stated in the manuscript. This was related to the validity of the findings. I think the authors should state this issue clearly in the “Methods” section and fully discuss its impact on the results.
Response: Thank you for your suggestion. All information in our study was self-reported by participants, which is now clearly stated in the Methods section. We also discussed its latent impact on the results.
“The following information was self-reported by participants: 1) socio-demographic information, including gender (male/female), age (in years), ethnicity (Han/others), college location (all cities in China were displayed for choose), education (junior college student or undergraduate student/graduate student), major (medical/non-medical) and family monthly income (<10000/10000-20000/>20000). 2) COVID-19 vaccination status (unvaccinated/vaccinated). 3) Reasons for not receiving first dose and booster vaccination (participants could choose at least one from the following options: contraindications/concerns about effectiveness/fear of adverse reactions/being too busy to attend to it/I think it is unnecessary/inconvenience; a blank was left to fill in for any other reason not mentioned above) were also collected.” (Line 91, Page 2)
“Finally, the vaccination status was self-reported by respondents, which may not be as accurate as clinical vaccination records. However, since personal vaccination records can be easily found via smart-phone in China and the vaccination information was required to be reported in details, information bias in this study is unlikely to be significant.” (Line 267, Page 8)
2. Were there any previous studies reported the status of COVID-19 vaccination in college students and its correlate? What are the main correlates of vaccination hesitancy in these previous studies? Please state in the introduction.
Response: Thank you for your constructive suggestion. We have additionally and specifically stated the key background information mentioned above in the Introduction section.
“Previous studies found vaccine hesitancy among college students was associated with socio-demographics such as gender, age [15], college location [13] and major [12]. Concerns about side effect, effectiveness also contributes to hesitancy [15]. A few studies reported college students’ actual vaccination status with the influence of vaccine hesitancy. Xiong et al. found nearly half of college students with vaccine hesitancy would still get vaccinated [12], which may partially account for the high coverage (over 95%) of first dose vaccination among them [16]. Cao [17] and Li [18] found the uptake of COVID-19 vaccination among college students was associated with gender, age and major, which was similar to vaccine hesitancy. However, with the promotion of COVID-19 booster vaccination, vaccine hesitancy may increase significantly [19].” (Line 56-64, Page 2)
Minor revision:
1. In “Abstract” section, when the format of the content in the first bracket is determined, the subsequent “AOR” and “95%CI” can be omitted.
Response: We have omitted the subsequent “AOR” and “95%CI” other than the first bracket.
2. Please note the format in “Characteristics of the participants” section (Line 122, Page3).
Response: Thanks for your reminder. We had carefully read, checked and revised the format in “Characteristics of the participants” section.
3. Please maintain the consistency of reporting all decimal places across the manuscript. P values should be expressed to 2 digits to the right of the decimal point unless the first 2 digits are zeros, in which case 3 digits to the right of the decimal place should be provided. However, values close to .05 may be reported to 3 decimal places because the .05 is an arbitrary cut point for statistical significance.
Response: Thank you for your reminder. We have maintained the consistency of reporting all decimal places across the manuscript.
4. There should be no lines below the total row in Table 2 and Table 3, and the lines between bivariate and multivariate analysis should break.
Response: We have revised the format of the tables according to IJERPH’ guideline for authors. The lines between bivariate and multivariate analysis are broken now.
Reviewer 2 Report
Authors performed a web-based survey to estimate the COVID-19 vaccination and its associated factors among Chinese college students. The paper is well written and easy to follow. Some minor issues should be resolved.
Methods:
The characteristics of the questionnaire (type of questions, closed or open, multiple answers etc) should be described in more detail.
Page 3 line 75: What does it mean "All vaccinated participants were included and classified according to the COVID-19 vaccination status." How % of non-vaccinated students was assessed?
How variables were selected for multivariable logistic regression?
Results:
4613 questionnaires seem to be quite a small number for the national study in China.
Discussion:
Please refer to students who completed the questionnaire as respondents, since only 3961 of all students in China completed the survey. Are there some national reports published in English to compare your results with the appropriate age range?
Line 189: please provide % of the vaccinated students for reference 19, since the website is only in Chinese.
Author Response
Thank you very much for giving us the opportunity to revise our manuscript. We sincerely appreciated the valuable comments and suggestions. We have carefully revised our manuscript according to your suggestions in a point-to-point manner with track changes. We believe that the quality of our manuscript has been substantially improved after addressing these edits. The main corrections and responses are as follows:
Methods:
1. The characteristics of the questionnaire (type of questions, closed or open, multiple answers etc) should be described in more detail.
Response: Thank you for your suggestion. We have added more details of the characteristics of the questionnaire in the Measure section.
“The following information was self-reported by participants: 1) socio-demographic information, including gender (male/female), age (in years), ethnicity (Han/others), college location (all cities in China were displayed for choose), education (junior college student or undergraduate student/graduate student), major (medical/non-medical) and family monthly income (<10000/10000-20000/>20000). 2) COVID-19 vaccination status (unvaccinated/vaccinated). 3) Reasons for not receiving first dose and booster vaccination (participants could choose at least one from the following options: contraindications/concerns about effectiveness/fear of adverse reactions/being too busy to attend to it/I think it is unnecessary/inconvenience; a blank was left to fill in for any other reason not mentioned above) were also collected.” (Line 91, Page 2)
2. Page 3 line 75: What does it mean "All vaccinated participants were included and classified according to the COVID-19 vaccination status." How % of non-vaccinated students was assessed?
Response: Thank you for your question. Here is our consideration about this issue. First, according to data released by the health authority in China, students who were only accounted for less than 5% of the total. In our study, non-vaccinated students only accounted for 0.5% (20 of 3916). Given the promotion of booster dose and high coverage of first dose, our study focused on the adherence to vaccination guideline after receiving vaccination, which we suppose is more meaningful. Second, the coverage of complete vaccination refers to the proportion of college students who have completed vaccination among all vaccinated college students. In order to reflect college students’ compliance with multi-dose vaccination, we used vaccinated students as the population to explore factors associated with complete vaccination. Third, if non-vaccinated college students were included in the analysis, they would be difficult to classify. On the one hand, it could be inappropriate to simply classify non-vaccinated students as students who received vaccination but have not completed it, because they were totally different groups. On the other hand, if they were classified as a separate group and when multinomial logistics regression was used, insufficient sample size may affect the robustness of model.
3. How variables were selected for multivariable logistic regression?
Response: Thank you for your suggestion. We have described this issue in detail in the Methods section.
“Variables with a p-value less than 0.10 in the bivariate analyses were introduced in multivariable logistic regression models. All effects were estimated with 95% CI and P values.” (Line 126, Page 3)
Results:
1. 4613 questionnaires seem to be quite a small number for the national study in China.
Response: For China, a country with nearly 20% of the world’s population, the sample size in our study does appear to be small. However, our actual sample size met the minimum estimated sample size. Moreover, respondents in our study were from different cities and covered all regions of the country. Compared with national statistical report about college students released by Chinese Ministry of Education in 2021 (the latest version), although there is a slight difference of sex ratio in our study, socio-demographics such as college location and educational level were consistent with the national report, which seems to be representative enough to show the real-world data. Additionally, the slight difference may result from convenience sampling, which has been mentioned in the Discussion section. Finally, the sample size in our study is similar to other similar nationwide studies, ranging from 3047 to 5641.
Discussion:
1. Please refer to students who completed the questionnaire as respondents, since only 3961 of all students in China completed the survey. Are there some national reports published in English to compare your results with the appropriate age range?
Response: We have changed the reference to students who completed the questionnaire. To our knowledge, the number of national reports about COVID-19 vaccination among college students was limited. Existing reports rarely touched on the information of booster vaccination. The latest report with the appropriate age range was about coverage of first-dose vaccination among Chinese students > 18 years of age, which was cited by the original version of our manuscript. However, given the diversity of reader population, we have replaced the original website with its English version.
2. Line 189: please provide % of the vaccinated students for reference 19, since the website is only in Chinese
Response: Thank you for your kind reminder. We’ve now provided the percentage of Chinese vaccinated students released by the Chinese government, and replaced the original website with its English version.
Reviewer 3 Report
This questionary study summarizes the opinion of young Chinese people on COVID-19 Vaccination.
The authors used clear and understandable language, and the flow of the paper was organized well. Material and Method are adequate. Presentation of results is convenient with charts and tables. The discussion section includes the results and limitations of the research.
My only contribution is to add the analysis of the information concerning the monthly income.
The author should pay attention to each table not to be displayed spreading over pages.
Author Response
Thank you very much for giving us the opportunity to revise our manuscript. We sincerely appreciated the valuable comments and suggestions. We have carefully revised our manuscript according to your suggestions in a point-to-point manner with track changes. We believe that the quality of our manuscript has been substantially improved after addressing these edits. The main corrections and responses are as follows:
1. My only contribution is to add the analysis of the information concerning the monthly income. The author should pay attention to each table not to be displayed spreading over pages.
Response: Thank you for your kind suggestion. We have added the analysis of the information concerning the monthly income. After adjustments, none of the tables was displayed spreading over pages.
“Moreover, similar to previous studies [12,15], family income level was not associated with vaccination behavior in our logistic regression model. Since COVID-19 vaccination is free of charge in China, concerns about vaccine prices would not be an objective barrier to vaccination for Chinese student.” (Line 238, Page 8)
Reviewer 4 Report
The article reported the coverage rate of COVID-19 vaccination among college students in China, focusing on the booster dose. No many studies explored this specific element in China. The demographic factors associated with the uptake and the reasons for refusal are well explained and the paper is well structured.
I suggest some minor revisions in the attachment, mainly to expand the introduction reporting the main reasons of hesitancy/refusal for COVID-19 vaccination reported in other studies.

Author Response
Thank you very much for giving us the opportunity to revise our manuscript. We sincerely appreciated the valuable comments and suggestions. We have carefully revised our manuscript according to your suggestions in a point-to-point manner with track changes. We believe that the quality of our manuscript has been substantially improved after addressing these edits. The main corrections and responses are as follows:
1. I suggest some minor revisions in the attachment, mainly to expand the introduction reporting the main reasons of hesitancy/refusal for COVID-19 vaccination reported in other studies.
Response: Thank you for your constructive suggestion. We think this part of background information is important to our study, and additionally and specifically stated it in the Introduction section.
“Previous studies found vaccine hesitancy among college students was associated with socio-demographics such as gender, age [15], college location [13] and major [12]. Concerns about side effect, effectiveness also contributes to hesitancy [15]. A few studies reported college students’ actual vaccination status with the influence of vaccine hesitancy. Xiong et al. found nearly half of college students with vaccine hesitancy would still get vaccinated [12], which may partially account for the high coverage (over 95%) of first dose vaccination among them [16]. Cao [17] and Li [18] found the uptake of COVID-19 vaccination among college students was associated with gender, age and major, which was similar to vaccine hesitancy. However, with the promotion of COVID-19 booster vaccination, vaccine hesitancy may increase significantly [19].” (Line 56-64, Page 2)